# Temporally Efficient Deep Learning with Spikes

**Peter O'Connor, Efstratios Gavves, Matthias Reisser, Max Welling**
QUVA Lab
University of Amsterdam
Amsterdam, Netherlands
`{p.e.oconnor,egavves,m.reisser,m.welling}@uva.nl`

## Abstract

The vast majority of natural sensory data is temporally redundant. For instance, video frames or audio samples which are sampled at nearby points in time tend to have similar values. Typically, deep learning algorithms take no advantage of this redundancy to reduce computations. This can be an obscene waste of energy. We present a variant on backpropagation for neural networks in which computation scales with the rate of change of the data - not the rate at which we process the data. We do this by implementing a form of Predictive Coding wherein neurons communicate a combination of their state, and their temporal change in state, and quantize this signal using Sigma-Delta modulation. Intriguingly, this simple communication rule give rise to units that resemble biologically-inspired leaky integrate-and-fire neurons, and to a spike-timing-dependent weight-update similar to Spike-Timing Dependent Plasticity (STDP), a synaptic learning rule observed in the brain. We demonstrate that on MNIST, on a temporal variant of MNIST, and on Youtube-BB, a dataset with videos in the wild, our algorithm performs about as well as a standard deep network trained with backpropagation, despite only communicating discrete values between layers.

## 1 Introduction

Currently, most algorithms used in Machine Learning work under the assumption that data points are independent and identically distributed, as this assumption provides good statistical guarantees for convergence. This is very different from the way data enters our brains. Our eyes receive a single, never-ending stream of temporally correlated data. We get to use this data once, and then it's gone. Moreover, most sensors produce sequential, temporally redundant streams of data. This can be both a blessing and a curse. From a statistical learning point of view this redundancy may lead to biased estimators when used to train models which assume independent and identically distributed input data. However, the temporal redundancy also implies that intuitively not all computations are necessary.

Online Learning is the study of how to learn in this domain - where data becomes available in sequential order and is given to the model only once. Given the enormous amount of sequential data, mainly videos, that are being produced nowadays, it seems desirable to develop learning systems that simply consume data on-the-fly as it is being generated, rather than collect it into datasets for offline-training. There is, however a problem of efficiency, which we hope to illustrate with two examples:

1. *CCTV feeds.* CCTV Cameras collect an enormous amount of data from mostly-static scenes. The amount of new information in a frame, given the previous frame, tends to be low, *i.e.* the data tends to be temporally redundant. If we want to train a model from of this data (for example a pedestrian detector), we need to process a large amount of mostly-static frames. If the frame rate doubles, so does the amount of computation. Intuitively, it feels that this should not be necessary. It would be nice to still be able to use all this data, but have the amount of computation scale with the amount of new information in each frame, not just the number of frames and dimensions of the data.

2. *Robot perception*. Robots have no choice but to learn online - their future input data (*e.g.* camera frames) are dependent on their previous predictions (*i.e.* motor actions). Not only does their data come in nonstationary temporal streams, but it typically comes from several sensors running at different rates. The camera may produce 1MB images at 30 frames/s, while the gyroscope might produce 1-byte readings at 1000 frames/s. It is not obvious, using current methods in deep learning, how we can integrate asynchronous sensory signals into a unified, trainable, latent representation, without undergoing the inefficient process of recomputing the function of the network every time a new signal arrives.

These examples point to the need for a training method where the amount of computation required to update the model scales with the amount of new information in the data, and not just the dimensionality of the data.

There has been a lot of work on increasing the computational efficiency of neural networks by quantizing neural weights or activations (see Section 4), but comparatively little work on exploiting redundancies in the data to reduce the amount of computation. O'Connor and Welling (2016b), set out to exploit the temporal redundancy in video by having neurons only send their quantized *changes* in activation to downstream neurons, and having the downstream neurons integrate these changes over time. This approach (take the temporal difference, multiply by weights, temporally integrate) works for efficiently approximating the function of the network, but fails for training. The reason for this failure is that when the weights are functions of time, we no longer reconstruct the correct activation for the next layer. In other words, given a sequence of inputs $x_0...x_t$ with $x_0 = 0$ and weights $w_1...w_t$: $\sum_{\tau=1}^{t}(x_\tau - x_{\tau-1}) \cdot w_\tau \neq x_t \cdot w_t$ unless $w_t = w_0 \forall t$. Figure 2 describes the problem visually.

In this paper, we correct for this problem by encoding a mixture of two components of the layers activation $x_t$: the *proportional* component $k_p x_t$, and the *derivative* component $k_d(x_t - x_{t-1})$. When we invert this encoding scheme, we get a decoding scheme which corresponds to taking an exponentially decaying temporal average of past inputs. Interestingly, the resulting neurons begin to resemble models of biological spiking neurons, whose membrane potentials can approximately be modeled as an exponentially decaying temporal average of past inputs.

In this work, we present a scheme wherein the temporal redundancy of input data is used to reduce the computation required to train a neural network. We demonstrate this on the MNIST and Youtube-BB datasets. To our knowledge we are the first to create a neural network training algorithm which uses less computation as data becomes more temporally redundant.

## 2 METHODS

We propose a coding scheme where neurons can represent their activations as a temporally sparse series of impulses. The impulses from a given neuron encode a combination of the value and the rate of change of the neuron's activation.

While our algorithm is designed to work efficiently with *temporal data*, we do not aim to learn *temporal sequences* in this work. We aim to efficiently approximate a function $y_t = f(x_t)$, where the current target $y_t$ is solely a function of the current input $x_t$, and not previous inputs $x_0...x_{t-1}$. The temporal redundancy between neighbouring inputs $x_{t-1}, x_t$ will however be used to make our approximate computation of this function more efficient.

### 2.1 PRELIMINARY

Throughout this paper we will use the notation $(f_3 \circ f_2 \circ f_1)(x) = f_3(f_2(f_1(x)))$ to denote function composition. We slightly abuse the notion of functions by allowing them to have an internal state which persists between calls. For example, we define the $\Delta$ function in Equation 1 as being the difference between the inputs in two consecutive calls (where persistent variable $x_{last}$ is initialized to 0). The $\Sigma$ function, defined in Equation 2, returns a running sum of the inputs over calls. So we can write, for example, that when our composition of functions $(\Sigma \circ \Delta)$ is called with a sequence of input variables $x_1...x_t$, then $(\Sigma \circ \Delta)(x_t) = x_t$, because $(x_1 - x_0) + (x_2 - x_1) + ... + (x_t - x_{t-1})|_{x_0=0} = x_t$.

In general, when we write $y_t = f(x_t)$, where $f$ is a function with persistent state, it will be implied that we have previously called $f(x_\tau)$ for $\tau \in [1,..,t-1]$ in sequence. Variable definitions that are used later will be highlighted in blue. While all terms are defined in the paper, we encourage the reader to refer to Appendix A for a complete collection of definitions and notations.

## 2.2 POSITION-DERIVATIVE (PD) ENCODING

Suppose a neuron has time-varying activation $x_1..x_t$. Taking inspiration from Proportional-Integral-Derivative (PID) controllers, we can "encode" this activation at each time step as a combination of its current activation and change in activation as $a_t \triangleq enc(x_t) = k_p x_t + k_d(x_t - x_{t-1})$, (see Equation 4). The parameters $k_p$ and $k_d$ determine what portion of our encoding represents the value of the activation and the rate of change of that value, respectively. In Section 4, we discuss how this is a form of *Predictive Coding* and in Appendix E, we discuss the effect our choices for these parameters have on the network.

To derive our decoding formula, we can simply solve for $x_t$ as $x_t = \frac{a_t + k_d x_{t-1}}{k_p + k_d}$ (Equation 5), such that $(dec \circ enc)(x_t) = x_t$. Notice that Equation 5 corresponds to decaying the previous decoder state by some constant $k_d/(k_p + k_d)$ followed by adding the input $a_t/(k_p + k_d)$. We can expand this recursively to see that this corresponds to a temporal convolution $a * \kappa$ where $\kappa$ is a causal exponential kernel $\kappa_\tau = \left\{ \frac{1}{k_p + k_d} \left( \frac{k_d}{k_d + k_p} \right)^\tau \text{ if } \tau \geq 0 \text{ otherwise } 0 \right\}$.

$$\Delta : x \mapsto y; \quad \text{Persistent: } x_{last} \leftarrow 0$$
$$y \leftarrow x - x_{last} \tag{1}$$
$$x_{last} \leftarrow x$$

$$\Sigma : x \mapsto y; \quad \text{Persistent: } y \leftarrow 0$$
$$y \leftarrow y + x \tag{2}$$

$$Q : x \mapsto y; \quad \text{Persistent: } \phi \leftarrow 0$$
$$\phi' \leftarrow \phi + x$$
$$y \leftarrow round(\phi') \tag{3}$$
$$\phi \leftarrow \phi' - y$$

$$enc : x \mapsto y; \quad \text{Persistent: } x_{last} \leftarrow 0$$
$$y \leftarrow k_p x + k_d(x - x_{last}) \tag{4}$$
$$x_{last} \leftarrow x$$

$$dec : x \mapsto y; \quad \text{Persistent: } y \leftarrow 0$$
$$y \leftarrow \frac{x + k_d y}{k_p + k_d} \tag{5}$$

$$R : x \mapsto round(x) \tag{6}$$

## 2.3 QUANTIZATION

Our motivation for the aforementioned encoding scheme is that we want a sparse signal which can be quantized into a low bitrate discrete signal. This will later be used to reduce computation. We can quantize our signal $a_t$ into a sparse, integer signal $s_t \triangleq Q(a_t)$, where the quantizer Q is defined in Equation 3. Equation 3 implements a form of Sigma-Delta modulation, a method widely used in signal processing to approximately communicate signals at low bit rates (Candy and Temes, 1962). We can show that $Q(x_t) = (\Delta \circ R \circ \Sigma)(x_t)$ (see Supplementary Material Section C), where $\Delta \circ R \circ \Sigma$ indicates applying a temporal summation, a rounding, and a temporal difference, in series. If $x_t$ is temporally redundant and we set $k_p$ to be small, then $|a_t| \ll 1 \forall t$, and we can expect $s_t$ to consist of mostly zeros with a few 1's and -1's.

We can now approximately reconstruct our original signal $x_t$ as $\hat{x}_t \triangleq dec(s_t)$ by applying our decoder, as defined in Equation 5. As our coefficients $k_p, k_d$ become larger, our reconstructed signal $\hat{x}_t$ should become closer to the original signal $x_t$. We illustrate examples of encoded signals and their reconstructions for different $k_p, k_d$ in Figure 1.

### 2.3.1 SPECIAL CASES

We can compactly write the entire reconstruction function as $\hat{x} = (dec \circ \Delta \circ R \circ \Sigma \circ enc)(x_t)$.

$k_p = 0$: When $k_p = 0$, we get $dec(x_t) = (k_d^{-1} \circ \Sigma)(x_t)$ and $enc(x_t) = (k_d \circ \Delta)(x_t)$, so our reconstruction reduces to $\hat{x} = (k_d^{-1} \circ \Sigma \circ \Delta \circ R \circ \Sigma \circ k_d \circ \Delta)(x_t)$. Because $\Sigma \circ k_d \circ \Delta$ all commute with one another, we can simplify this to $\hat{x}_t = (k_d^{-1} \circ R \circ k_d)(x_t)$. so our decoded signal is

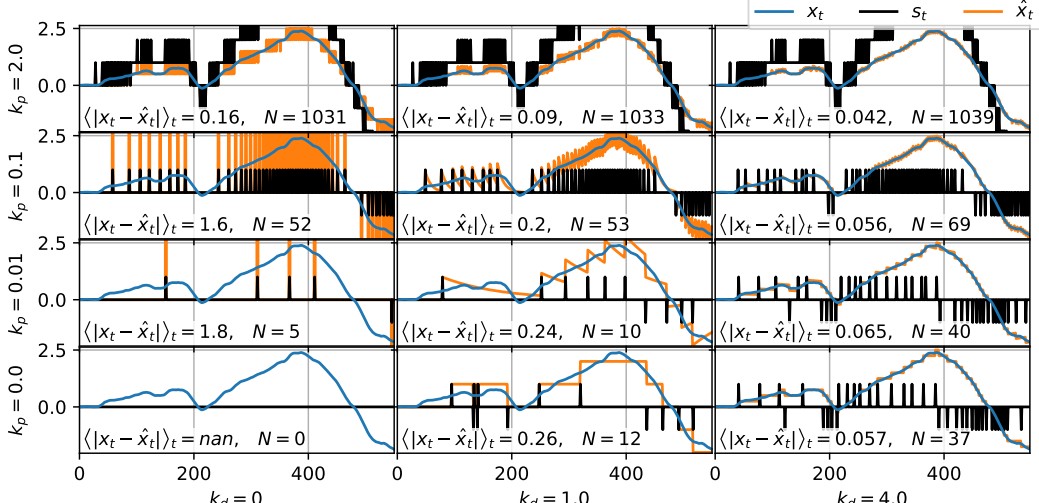

Figure 1: An example signal $x_t$ (blue), encoded with $k_p$ varying across rows and $k_d$ varying across columns. $s_t$ (black) is the quantized signal produced by the successive application of encoding (Equation 4) and quantization (Equation 3, where $N$ indicates the total number of spikes. $\hat{x}_t$ (orange) is the reconstruction of $x_t$ produced by applying Equation 5 to $s_t$. One might, after a careful look at this figure, ask why we bother with the proportional ($k_p$) term at all? Figure 2 anticipates this question and answers it visually.

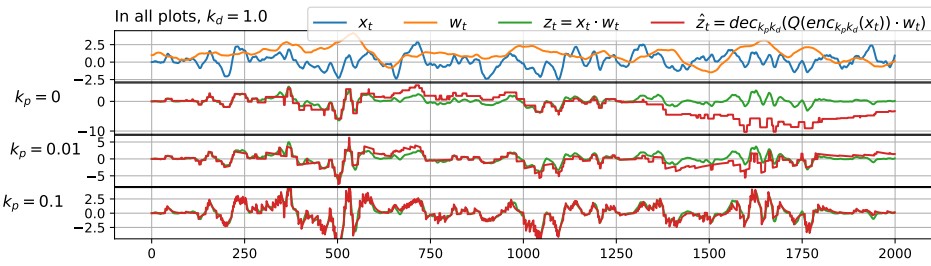

Figure 2: The problem with *only* sending changes in activation (i.e. $k_p = 0$) is that during training, weights change over time. **Top:** we generate random signals for a single scalar activation $x_t$ and scalar weight $w_t$. **Row 2:** We efficiently approximate $z_t$ by taking the temporal difference, multiplying by $w_t$ then temporally integrating, to produce $\hat{z}_t$, as described in Section 2.4. As the weight $w_t$ changes over time, our estimate $\hat{z}$ diverges from the correct value. **Rows 3, 4:** Introducing $k_p$ allows us to bring our reconstruction back in line with the correct signal.

$\hat{x}_t = round(x_t \cdot k_d)/k_d$, with no dependence on $x_{t-1}$. This is visible in the bottom row of Figure 1. This was the encoding scheme used in O'Connor and Welling (2016b).

$k_d = 0$: In this case, $dec(x_t) = k_p^{-1} x_t$ and $enc(x_t) = k_p x_t$ so our encoding-decoding process becomes $\hat{x} = (k_p^{-1} \circ \Delta \circ R \circ \Sigma \circ k_p)(x_t)$. Neither our encoder nor our decoder have any memory, and we take no advantage of temporal redundancy.

## 2.4 SPARSE COMMUNICATION BETWEEN LAYERS

The purpose of our encoding scheme is to reduce computation by sparsifying communication between layers of a neural network. Our approach is to approximate the matrix-product as a series of additions, where the number of additions is inversely proportional to the sparsity of the input data. Suppose we

are trying to compute the pre-nonlinearity activation of the first hidden layer, $z_t \in \mathbb{R}^{d_{out}}$, given the input activation, $x_t \in \mathbb{R}^{d_{in}}$. We approximate $z_t$ as:

$$z_t \triangleq x_t \cdot w_t \approx \hat{x}_t \cdot w_t \triangleq dec(Q(enc(x_t))) \cdot w_t \triangleq dec(s_t) \cdot w_t \approx dec(s_t \cdot w_t) \triangleq \hat{z}_t$$
$$\text{where: } x_t, \hat{x}_t \in \mathbb{R}^{d_{in}}; s_t \in \mathbb{Z}^{d_{in}}; w \in \mathbb{R}^{d_{in} \times d_{out}}; z_t, \hat{z}_t \in \mathbb{R}^{d_{out}} \quad (7)$$

The first approximation comes from the quantization (Q) of the encoded signal, and the second from the fact that the weights change over time, as explained in Figure 2. The effects of these approximations are further explored in Appendix E.1.

Computing $z_t$ takes $d_{in} \cdot d_{out}$ multiplications and $(d_{in} - 1) \cdot d_{out}$ additions. The cost of computing $\hat{z}_t$, on the other hand, depends on the contents of $s_t$. If the data is temporally redundant, $s_t \in \mathbb{Z}^{d_{in}}$ should be sparse, with total magnitude $N \triangleq \sum_i |s_{t,i}|$. $s_t$ can be decomposed into a sum of one-hot vectors $s_t = \sum_{n=1}^{N} sign(s_{t,i_n}) \cdot \gamma_{i_n} : i_n \in [1..d_{i_n}]$ where $\gamma_{i_n}$ is a length-$d_{in}$ one-hot vector with element $(\gamma_{i_n})_{i_n} = 1$. The matrix product $s_t \cdot w$ can then be decomposed into a series of row additions:

$$s_t \cdot w = \left( \sum_{n=1}^{N} \text{sign}(s_{t,i_n}) \cdot \gamma_{i_n} \right) \cdot w = \sum_{n=1}^{N} \text{sign}(s_{t,i_n}) \gamma_{i_n} \cdot w = \sum_{n=1}^{N} \text{sign}(s_{t,i_n}) \cdot w_{i_n,}. \quad (8)$$

If we include the encoding, quantization, and decoding operations, our matrix product takes a total of $2d_{in} + 2d_{out}$ multiplications, and $\sum_n |s_{t,n}| \cdot d_{out} + 3d_{in} + d_{out}$ additions. Assuming the $\sum_n |s_{t,n}| \cdot d_{out}$ term dominates, we can say that the relative cost of computing $\hat{z}_t$ vs $z_t$ is:

$$\frac{cost(\hat{z})}{cost(z)} \approx \frac{\sum_n |s_{t,n}| \cdot cost(add)}{d_{in} \cdot (cost(add) + cost(mult))} \quad (9)$$

## 2.5 A Neural Network

We can implement this encoding scheme on every layer of a neural network. Given a standard neural net $f_{nn}$ consisting of alternating linear ($\cdot w_l$) and nonlinear ($h_l$) operations, our network function $f_{pdnn}$ can then be written as:

$$f_{nn}(x) = (h_L \circ \cdot w_L \circ ... \circ h_1 \circ \cdot w_1)(x) \quad (10)$$
$$f_{pdnn}(x) = (h_L \circ dec_L \circ w_L \circ Q_L \circ enc_L \circ ... \circ h_1 \circ dec_1 \circ \cdot w_1 \circ Q_1 \circ enc_1)(x) \quad (11)$$

We can use the same approach to approximately calculate our gradients to use in training. If we define our layer activations as $\hat{z}_l \triangleq (dec \circ \cdot w_l \circ Q \circ enc)(x)$ if $l = 1$ otherwise $(dec \circ \cdot w_l \circ Q \circ enc)(\hat{z}_{l-1})$, and $\mathcal{L} \triangleq \ell(f_{pdnn}(x), y)$, where $\ell$ is some loss function and $y$ is a target, we can backpropagate the approximate gradients as:

$$\widehat{\frac{\partial \mathcal{L}}{\partial \hat{z}_l}} = \begin{cases} \frac{\partial \mathcal{L}}{\partial z_L} & \text{if } l = L \\ \left( \odot h_l'(\hat{z}_l) \circ dec_l^{back} \circ \cdot w_{l+1}^T \circ Q_{l+1}^{back} \circ enc_{l+1}^{back} \right) \left( \widehat{\frac{\partial \mathcal{L}}{\partial \hat{z}_{l+1}}} \right) & \text{otherwise} \end{cases} \quad (12)$$

This can be implemented by either executing a (sparse) forward and backward pass at each time-step, or in an "event-based" manner, where the quantizers fire "events" whenever incoming events push their activations past a threshold, and these events are in turn sent to downstream neurons. For ease of implementation, we opt for the former in our code. Note that unlike in regular backprop, computing these forward and backward passes results in changes to the internal state of the $enc$, $dec$, and $Q$ components.

## 2.6 Parameter Updates

There is no use in having an efficient backward pass if the parameter updates are not also efficient. In a normal neural network trained with backpropagation and simple stochastic gradient descent, the

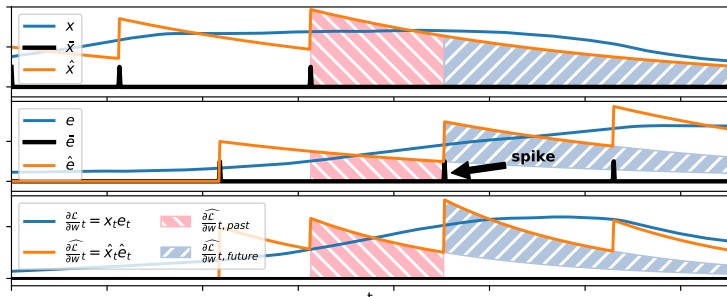

Figure 3: A visualization of our efficient update schemes from Section 2.6. **Top**: A scalar signal representing a presynaptic neuron activation $x_t = h_{l-1}(z_l - 1)$, its quantized version, $\bar{x}_t = (Q \circ enc)(x_t)$, and its reconstruction $\hat{x}_t = dec(\bar{x}_t)$. **Middle**: Another signal, representing the postsynaptic gradient of the error $e = \frac{\partial \mathcal{L}}{\partial z_l}$, along with its quantized ($\bar{e}$) and reconstructed ($\hat{e}$) variants. **Bottom**: The true weight gradient $\frac{\partial \mathcal{L}}{\partial w_t}$ and the reconstruction gradient $\widehat{\frac{\partial \mathcal{L}}{\partial w_t}}$. At the time of the spike in $\bar{e}_t$, we have two schemes for efficiently computing the weight gradient that will be used to increment weight (see Section 2.6). The *past* scheme computes the area under $\hat{x} \cdot \hat{e}$ since the last spike, and the *future* scheme computes the total future additional area due to the current spike.

parameter update for weight matrix $w$ has the form $w \leftarrow w - \eta \frac{\partial \mathcal{L}}{\partial w}$ where $\eta$ is the learning rate. If $w$ connects layer $l - 1$ to layer $l$, we can write $\frac{\partial \mathcal{L}}{\partial w} = x_t \otimes e_t$ where $x_t \triangleq h_{l-1}(z_{l-1,t}) \in \mathbb{R}^{d_{in}}$ is the presynaptic (layer $l - 1$) activation, $e_t \triangleq \frac{\partial \mathcal{L}}{\partial z_{l,t}} \in \mathbb{R}^{d_{out}}$ is the postsynaptic (layer $l$) backpropagating gradient and $\otimes$ is the outer product. So we require $d_{in} \cdot d_{out}$ multiplications to update the parameters for each sample.

We want a more efficient way to compute this product, which takes advantage of the sparsity of our encoded signals to reduce computation. We can start by applying our encoding-quantizing-decoding scheme to our input and error signals as $\bar{x}_t \triangleq (Q \circ enc)(x_t) \in \mathbb{Z}^{d_{in}}$ and $\bar{e}_t \triangleq (Q \circ enc)(e_t) \in \mathbb{Z}^{d_{out}}$, and approximate our true update as $\widehat{\frac{\partial \mathcal{L}}{\partial w}}_{recon,t} \triangleq \hat{x}_t \otimes \hat{e}_t$ where $\hat{x}_t \triangleq dec(\bar{x}_t)$ and $\hat{e}_t \triangleq dec(\bar{e}_t)$. This does not do any good by itself, because the vectors involved in the outer product, $\hat{x}_t$ and $\hat{e}_t$, are still not sparse. However, we can exactly compute the sum of this value over time using one of two sparse update schemes - *past updates* and *future updates* - which are depicted in Figure 3. We give the formula for the Past and Future update rules in Appendix D, but summarize them here:

**Past Updates**: For a given synapse $w_{i,j}$, if either the presynaptic neuron spikes ($\bar{x}_{t_i} \neq 0$) or the postsynaptic neuron spikes ($\bar{e}_{t_i} \neq 0$), we increment the $w_{i,j}$ by the total area under $\hat{x}_{\tau,i}\hat{e}_{\tau,j}$ since the last spike. We can do this efficiently because between the current time and the time of the previous spike, $\hat{x}_{\tau,i}\hat{e}_{\tau,j}$ is a geometric sequence. Given a known initial value $u$, final value $v$, and decay rate $r$, a geometric sequence sums to $\frac{u-v}{1-r}$. The area calculated is shown in pink on the bottom row of Figure 3, and one algorithm to calculate it is in Equation 19 in Appendix D.

**Future Updates**: Another approach is to calculate the Present Value of the future area under the integral from the current spike. This is depicted in the blue-gray area in Figure 3, and the formula is in Equation 20 in Appendix D.

Finally, because the magnitude of our gradient varies greatly over training, we create a scheme for adaptively tuning our $k_p$, $k_d$ parameters to match the running average of the magnitude of the data. This is described in detail in Appendix E.

## 2.7 RELATION TO STDP

An extremely attentive reader might have noted that Equation 20 has the form of an online implementation of Spike-Timing Dependent Plasticity (STDP). STDP (Markram et al., 2012) emerged from neuroscience, where it was observed that synaptic weight changes appeared to be functions of the

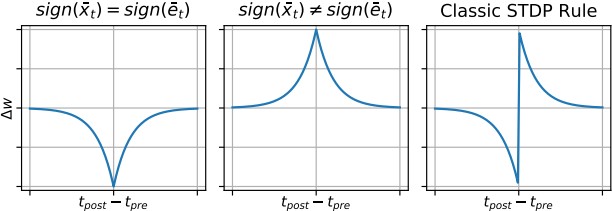

Figure 4: **Left**: Our STDP rule, when both the input and error spikes have the same sign. **Middle**: Our STDP rule, when the input and error spikes have opposite signs. **Right**: The classic STDP rule Markram et al. (2012), where the weight update is positive when a presynaptic spike preceeds a postsynaptic spike, and negative otherwise.

relative timing of pre- and post-synaptic spikes. The empirically observed function usually has the double-exponential form seen on the rightmost plot of Figure 4.

Using the quantized input signal $\bar{x}$ and error signal $\bar{e}$, and their reconstructions $\hat{x}_t$ and $\hat{e}_t$ as defined in the last section, we define a causal convolutional kernel $\kappa_t = \left\{ k_\beta (k_\alpha)^t \text{ if } t \geq 0 \text{ otherwise } 0 \right\}$, where $k_\alpha \triangleq= \frac{k_d}{k_p+k_d}$, $k_\beta \triangleq \frac{1}{k_p+k_d}$. We can then define a "cross-correlation kernel" $g_t = \left\{ \kappa_t \text{ if } t \geq 0 \text{ otherwise } \kappa_{-t} \right\} = k_\beta(k_\alpha)^{|t|} : t \in \mathbb{Z}$ which defines the magnitude of a parameter update as a function of the difference in timing between pre-synaptic spikes from the forward pass and post-synaptic spikes from the backward pass. The middle plot of Figure 4 is a plot of $g$. We define our STDP update rule as:

$$\widehat{\frac{\partial \mathcal{L}}{\partial w}}_{t,STDP} = \left( \sum_{\tau=-\infty}^{\infty} \bar{x}_{t-\tau} g_\tau \right) \otimes \bar{e}_t \tag{13}$$

We note that while our version of STDP has the same double-exponential form as the classic STDP rule observed in neuroscience (Markram et al., 2012), our "presynaptic" spikes come from the forward pass while our "postsynaptic" spikes come from the *backwards* pass. STDP is not normally used to as a learning rule networks trained by backpropagation, so the notion of forward and backward pass with a spike-timing-based learning rule are new. Moreover, unlike in classic STDP, we do not have the property that sign of the weight change depends on whether the presynaptic spike preceded the postsynaptic spike.

In Section D in the supplementary material we show experimentally that while Equations $\widehat{\frac{\partial \mathcal{L}}{\partial w}}_{recon}$, $\widehat{\frac{\partial \mathcal{L}}{\partial w}}_{past}$, $\widehat{\frac{\partial \mathcal{L}}{\partial w}}_{future}$, $\widehat{\frac{\partial \mathcal{L}}{\partial w}}_{stdp}$ may all result in different updates at different times, the rules are equivalent in that for a given set of pre/post-synaptic spikes $\bar{x}, \bar{e}$, the cumulative sum of their updates over time converges exactly.

## 3 EXPERIMENTS

### 3.1 TEMPORAL MNIST

To evaluate our network's ability to learn, we train it on the standard MNIST dataset, as well as a variant we created called "Temporal MNIST". Temporal MNIST is simply a reshuffling of the MNIST dataset so that so that similar inputs (in terms of L2-pixel distance), are put together. Figure 6 shows several snippets of consecutive frames in the temporal MNIST dataset. We compare our Proportional-Derivative Net against a conventional Multi-Layer Perceptron with the same architecture (one hidden layer of 200 ReLU hidden units and a softmax output). The results are shown in Figure 5. Somewhat surprisingly, our predictor slightly outperformed the MLP, getting 98.36% on the test set vs 98.25% for the MLP. We assume this improvement is due to the regularizing effect of the quantization. On Temporal MNIST, our network was able to converge with less computation than it required for MNIST (it used $32 \cdot 10^{12}$ operations for MNIST vs $15 \cdot 10^{12}$ for Temporal MNIST), but

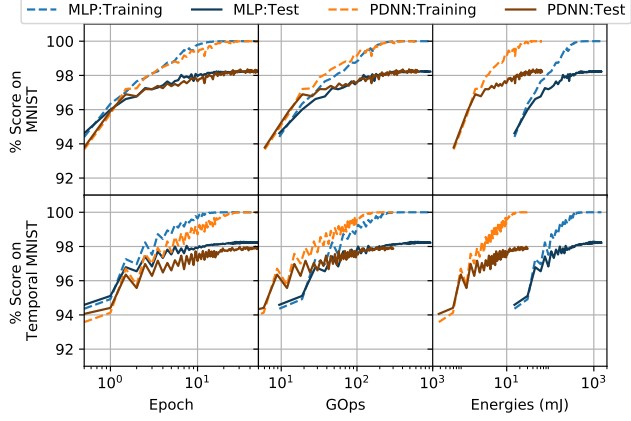

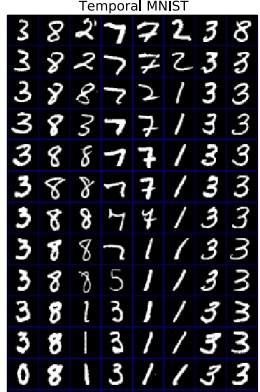

Figure 5: Top Row: Results on MNIST. Bottom Row: Results on Temporal MNIST. Left Column: the training and test scores as a function of epoch. Middle: We now put the number of computational operations on the x-axis. We see that as a result our PDNN shifts to the left. Right: Because our network computes primarily with additions rather than multiplications. When we multiply our operation counts with the estimates of Horowitz (2014) for the computational costs of arithmethic operations (0.1pJ for 32-bit fixed-point addition vs 3.2pJ for multiplication), we can see that our algorithm would be at an advantage on any hardware where arithmetic operations were the computational bottleneck.

Figure 6: Some samples from the Temporal-MNIST dataset. Each column shows a snippet of adjacent frames.

ended up with a slightly worse test score when compared with the MLP (the PDNN achieved 97.99% vs 98.28% for the MLP). The slightly higher performance of the MLP on Temporal MNIST may be explained by the fact that the gradients on Temporal MNIST tend to be correlated across time-steps, so weights will tend to move in a single direction for a number of steps, which will interfere with the PDNN's ability to accurately track layer activations (see Figure 2). Appendix F contains a table of results with varying hyperparameters.

## 3.2 YouTube Video Dataset

Next, we want to simulate the setting of CCTV cameras, discussed in Section 1, where we have a lot of data with only a small amount of new information per frame. In the absence of large enough public CCTV video datasets, we investigate the surrogate task of frame-based object classification on wild YouTube videos from the large, recently released Youtube-BB dataset Real et al. (2017). Our subset consists of 358 Training Videos and 89 Test videos with 758,033 frames in total. Each video is labeled with an object in one of 24 categories.

We start from a VGG19 network (Simonyan and Zisserman, 2014): a 19-layer convolutional network pre-trained on imagenet. We replace the top three layer with three of our own randomly initialized layers, and train the network both as a spiking network, and as a regular network with backpropagation. While training the entire spiking network end-to-end works, we choose to only train the top layers, in order to speed up our training time.

We compare our training scores and computation between a spiking and non-spiking implementation. The learning curves in Figure 7 show that our spiking network performs comparably to a non-spiking network, and Figure 8 shows how the computation per frame of our spiking network decreases as we increase the frame rate (i.e. as the input data becomes more temporally redundant). Because our spiking network uses only additions, while a regular deep network does multiply-adds, we use the estimated energy-costs per op of Horowitz (2014) to compare computations to a single scale, which estimates the amount of energy required to do multiplies and adds in fixed-point arithmetic.

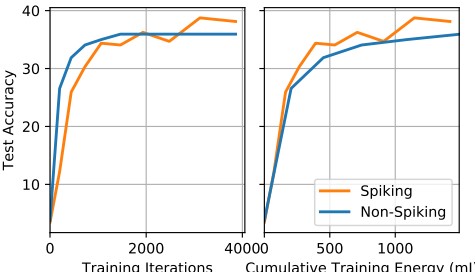 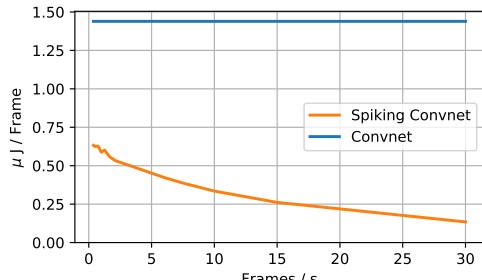

Figure 7: Left: Learning Curves on the Youtube Dataset. Right: Learning curves with respect to computational energy using the conversion of Horowitz (2014). The spiking network slightly outperforms the non-spiking baseline - we suspect that this is because the added noise of spiking acts as a regularizer.

Figure 8: We simulate different frame-rates by selecting every n'th frame. This plot shows our network's mean computation over several snippets of video, at varying frame rates. As our frame rate increases, the computation per-frame of our spiking network goes down, while with a normal network, it remains fixed.

## 4    RELATED WORK

Noise-Shaping is a quantization technique that aims to increase the fidelity of signal reconstructions, per unit of bandwidth of the encoded signal, by quantizing the signal in such a way that the quantization noise is pushed into a higher frequency band which is later filtered out upon decoding. Sigma-Delta (also known as Delta-Sigma) quantization is a form of noise-shaping. Shin (2001) first suggested that biological neurons may be performing a form of noise shaping, and Yoon (2017) found standard spiking neuron models actually implement a form of Sigma-Delta modulation.

The encoding/decoding scheme we use in this paper can be seen as a form of Predictive Coding. Predictive coding is a lossless compression technique wherein the predictable parts of a signal are subtracted away so that just the unpredictable parts are transmitted. The idea that biological neurons may be doing some form of predictive coding was first proposed by Srinivasan et al. (1982). In a predictive-coding neuron (unlike neurons commonly used in Deep Learning), there is a distinction between the signal that a neuron *represents* and the signal that it *transmits*. The neurons we use in this paper can be seen as implementing a simple form of predictive coding where the "prediction" is that the neuron maintains a decayed form of its previous signal - i.e. that $pred(x_t) \triangleq \frac{k_d}{k_p+k_d}x_{t-1}$ (See Appendix B for detail). Chklovskii and Soudry (2012) suggest that the biological spiking mechanism may be thought of as consisting of a sigma-delta modulator within a predictive-coding circuit.

To our knowledge, none of the aforementioned work has yet been used in the context of deep learning.

There has been sparse but interesting work on merging the notions of spiking neural networks and deep learning. Diehl et al. (2015) found a way to efficiently map a trained neural network onto a spiking network. Lee et al. (2016) devised a method for training integrate-and-fire spiking neurons with backpropagation - though their neurons did not send a temporal difference of their activations. O'Connor and Welling (2016a) created a method for training event-based neural networks - but their method took no advantage of temporal redundancy in the data. Binas et al. (2016) and (O'Connor and Welling, 2016b) both took the approach of sending quantized temporal changes to reduce computation on temporally redundant data, but their schemes could not be used to train a neural network. Bohte et al. (2000) showed how one could apply backpropagation for training spiking neural networks, but it was not obvious how to apply the method to non-spiking data. Zambrano and Bohte (2016) developed a spiking network with an adaptive scale of quantization (which bears some resemblance to our tuning scheme described in Appendix E), and show that the spiking mechanism is a form of Sigma-Delta modulation, which we also use here. Courbariaux et al. (2015) showed that neural networks could be trained with binary weights and activations (we just quantize activations). Bengio et al. (2015) found a connection between the classic STDP rule (Figure 4, right) and optimizing a dynamical neural network, although the way they arrived at an STDP-like rule was quite different from ours (they

frame STDP as a way to minimze an objective based on the rate of change of the real-valued state of the network, whereas we use it approximately compute gradients based on spike-encodings of layer activations).

## 5 DISCUSSION

We set out with the objective of reducing the computation in deep networks by taking advantage of temporal redundancy in data. We described a simple rule (Equation 4) for sparsifying the communication between layers of a neural network by having our neurons communicate a combination of their temporal change in activation, and the current value of their activation. We show that it follows from this scheme that neurons should behave as leaky integrators (Equation 5). When we quantize our neural activations with Sigma-Delta modulation, a common quantization scheme in signal processing, we get something resembling a leaky integrate-and-fire neuron. We derive efficient update rules for the weights of our network, and show these to be related to STDP - a learning rule first observed in neuroscience. Finally, we train our network, verify that it does indeed compute more efficiently on temporal data, and show that it performs about as well as a traditional deep network of the same architecture, but with significantly reduced computation. Finally, we show that our network can train on real video data.

The efficiency of our approach hinges on the temporal redundancy of our input data and neural activations. There is an interesting synergy here with the concept of slow-features (Wiskott, 1999). Slow-Feature learning aims to discover latent objects that persist over time. If the hidden units were to specifically learn to respond to slowly-varying features of the input, the layers in a spiking implementation of such a network would have to communicate less often. In such a network, the tasks of feature-learning and reducing inter-layer communication may be one and the same.

Code is available at github.com/petered/pdnn.

### ACKNOWLEDGMENTS

This work was supported by Qualcomm, who we'd also like to thank for sharing their past work with us.

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

## A    NOTATION

Here we present a legend of notation used throughout this paper. While the paper is intended to be self-contained, the reader may want to consult this list if ever there is any doubt about the meaning of a variable used in the paper. Here we indicate the section in which each symbol is first used.

**Section 2.1**

$\Delta$: A "temporal difference" operator. See Equation 1

$\Sigma$: A "temporal integration" operator. See Equation 2

$Q$: Sigma-Delta quantization. See Equation 3

$\phi$: The internal state variable of the quantizer $Q$.

$enc$: An "encoding" operation, which takes a signal and encodes it into a combination of the signal's current value and its change in value since the last time step. See Equation 4.

$dec$: A "decoding" operation, which takes an encoding signal and attempts to reconstruct the original signal that it was encoded from. If there was no quantization done on the encoded signal, the reconstruction will be exact, otherwise it will be an approximation. See Equation 5.

$R$: The "rounding" operation, which simply rounds an input to the nearest integer value.

$x_t$: Used throughout the paper to represent the value of a generic analog input signal at time $t$. In Sections 2.1, 2.2, and 2.3 it represents a scalar, and thereafter it represents a vector of inputs.

**Section 2.2**

$k_p, k_d \in \mathbb{R}^+$: Positive scalar coefficients used in the encoder and decoder, controlling how the extent to which the encoding is *proportional* to the input ($k_p$) vs *proportional to the temporal difference* of the input ($k_d$).

$a_t \triangleq enc(x_t)$: Used to represent the encoded version of $x_t$.

**Section 2.3**

$s_t \triangleq Q(a_t)$: Used to represent the quantized, encoded version of $x_t$.

$\hat{x}_t \triangleq dec(s_t)$: Used to represent the reconstruction of input $x_t$, obtained by encoding, quantizing, and decoding $x_t$.

**Section 2.4**

$w_t \in \mathbb{R}^{d_{in} \times d_{out}}$ is the value of a weight matrix at time $t$.

$z_t \triangleq x_t \cdot w_t \in \mathbb{R}^{d_{out}}$ is the value of a pre-nonlinearity hidden layer activation in a non-spiking network at time $t$.

$\hat{z}_t \triangleq dec(Q(enc(x_t)) \cdot w_t) \in \mathbb{R}^{d_{out}}$ is the value of a pre-nonlinearity hidden layer activation in the spiking network at time $t$. It is an approximation of $z_t$.

**Section 2.5**

$(\cdot w_l)$ indicates applying a function which takes the dot-product of the input with the $l$'th weight matrix: $(\cdot w_l)(x) \triangleq x \cdot w_l$

$h_l$ indicates an elementwise nonlinearity (e.g. a ReLU).

$Q_l$ indicates the quantization step applied at the $l$'th layer (because quantization has internal state, $\phi$, and an associated layer dimension, we use the subscript to distinguish quantizers at different layers.)

$dec_l, enc_l$ are likewise the (stateful) encoding/decoding functions applied before/after layer $l$.

$\frac{\partial \mathcal{L}}{\partial z_l}$ is the derivative of the loss with respect to the (pre-nonlinearity) activation of layer $l$.

$(\cdot w_l^T)$ indicates the dot product with the *transpose* of $w_l$. This is simply backpropagation across a weight matrix: If $u \triangleq x \cdot w_l$, then $\frac{\partial \mathcal{L}}{\partial x} = \frac{\partial \mathcal{L}}{\partial u} \cdot w_l^T$

$\hat{z}_l$ is the approximation to the (pre-nonlinearity) activation to layer $l$ (ie the output of $dec_l$), computed by the spiking network.

$(\odot h'_l(\hat{z}_l))$ is a function that performs an elementwise multiplication of the input with the derivative of nonlinearity $h_l$ evaluated at $\hat{z}_l$. This is simply backpropagation across a nonlinearity: If $u \triangleq h_l(x)$, then $\frac{\partial \mathcal{L}}{\partial x} = \frac{\partial \mathcal{L}}{\partial u} \odot h'_l(x)$

$dec_l^{back}$, $enc_l^{back}$, $Q_l^{back}$ serve the same functions as $dec_l$, $enc_l$, $Q_l$, but for the backward pass.

$\widehat{\frac{\partial \mathcal{L}}{\partial \hat{z}_l}} \in \mathbb{R}^{d_l}$ Is our approximation to the derivative of the loss of our network with respect to $\hat{z}_l$, which is itself an approximation of the activation $z_l$ in a non-spiking implementation of the network.

**Section 2.6**

In the updates section we describe how we calculate the weight gradients in layer $l$. Because this description holds for any arbitrary layer, we get rid of the layer subscript and use the following notation:

$x_t \triangleq h_{l-1}(z_{l-1,t}) \in \mathbb{R}^{d_{in}}$ here is defined as a shorthand for "the input to layer $l$".

$e_t \triangleq \widehat{\frac{\partial \mathcal{L}}{\partial \hat{z}_{l,t}}} \in \mathbb{R}^{d_{out}}$ is simply a shorthand for "the approximate backpropagated gradient at layer $l$"

$(\bar{x}_t$ and $\bar{e}_t)$ are the encoded and quantized versions of signals $(x_t$ and $e_t)$

$(\hat{x}_t$ and $\hat{e}_t)$ are the reconstructed versions of signals $(x_t$ and $e_t)$, taken from the quantized $(\bar{x}_t$ and $\bar{e}_t)$

$\widehat{\frac{\partial \mathcal{L}}{\partial w}}_{recon,t} \triangleq \hat{x}_t \otimes \hat{e}_t \in \mathbb{R}^{d_{in} \times d_{out}}$ is the approximate gradient of weight matrix $w$, as calculated by taking the outer product of the (input, error) reconstructions, $\hat{x}, \hat{e}$.

$\left( \widehat{\frac{\partial \mathcal{L}}{\partial w}}_{past,t}, \widehat{\frac{\partial \mathcal{L}}{\partial w}}_{future,t} \right)$ are the gradient approximations at time $t$ taken using the (past, future) approximation methods, defined in Appendix $D$. They are more efficient to calculate than $\widehat{\frac{\partial \mathcal{L}}{\partial w}}_{recon,t}$ but converge to the same value when averaged over time (i.e. $\lim_{T \to \infty} \frac{1}{T} \sum_t^T \widehat{\frac{\partial \mathcal{L}}{\partial w}}_{future,t} = \frac{1}{T} \sum_t^T \widehat{\frac{\partial \mathcal{L}}{\partial w}}_{recon,t}$ (see Figure 9).

**Section 2.7**

$\widehat{\frac{\partial \mathcal{L}}{\partial w}}_{stdp,t}$ is the gradient approximation taken using the STDP-type update. It also converges to the same value as $\widehat{\frac{\partial \mathcal{L}}{\partial w}}_{recon,t}$ when averaged out over time.

$k_\alpha \triangleq \frac{k_d}{k_p + k_d} \in [0, 1], k_\beta \triangleq \frac{1}{k_p + k_d} \in R^+$: A reparametrization of $k_p, k_d$ in terms of the *memory* in our decoder $k_\alpha$ and the *scaling of our encoded signal* ($k_\beta$). This reparametrization is also used when discussing the automatic tuning of $k_p, k_d$ to match the dynamic range of our data in Appendix E

# B  RELATION TO PREDICTIVE CODING

Our encoding/decoding scheme is an instance of predictive coding - an idea imported from the signal processing literature into neuroscience by Srinivasan et al. (1982) wherein the power of a transmitted signal is reduced by subtracting away the predictable component of this signal before transmission, then reconstructing it after (This requires that the encoder and decoder share the same prediction model).

Bharioke and Chklovskii (2015) formulate feedforward predictive coding as follows (with variables names changed to match the conventions of this paper):

$$a_t \triangleq x_t - C_{feedforward}(x_{t-1}, x_{t-2}, ...) \tag{14}$$

$$= x_t - \sum_{\tau=1}^{\infty} \omega_\tau x_{t-\tau} \qquad \text{In the case of Linear Predictive Coding} \tag{15}$$

Where the reconstruction is done by:

$$x_t = a_t + C_{feedforward}(x_{t-1}, x_{t-2}, ...) \tag{16}$$

They go on to define "optimal" liner filter parameters $[w_1, w_2, ...]$ that minimize the average magnitude of $a_t$ in terms of the autocorrelation and signal-to-noise ratio of $x$.

Our scheme defines:

$$a_t \triangleq k_p x_t + k_d(x_t - x_{t-1}) = (k_p + k_d)\left(x_t - \frac{k_d}{k_p + k_d}x_{t-1}\right) \tag{17}$$

So it is identical to feedforward predictive coding with $\omega_\tau = \begin{cases} \frac{k_d}{k_p+k_d} & \text{if } \tau = 1 \\ 0 & \text{otherwise} \end{cases}$ up to a scaling constant of $(k_p + k_d)$. In our case, the function of this additional constant is to determine the coarseness of the quantization.

From this relationship it is clear that this work could be extended to come up with more efficient predictive coding schemes which could further reduce computation by learning the temporal characteristics of the input signal.

## C  SIGMA-DELTA UNWRAPPING

Here we show that $Q = \Delta \circ R \circ \Sigma$, where $Q, \Delta, R, \Sigma$ are defined in Equations 3, 2, 6, 1, respectively.

From Equation 3 (Q) we can see that

$$y_t \leftarrow round(x_t + \phi_{t-1}) \in \mathbb{Z}$$
$$\phi_t \leftarrow \phi_{t-1} + x_t - y_t \in \mathbb{R}$$

Now we can unroll for $y_t$ and use the fact that if $s \in \mathbb{Z}$ then $round(a + s) = round(a) + s$:

$$
\begin{aligned}
y_t &= round(x_t + \phi_{t-1}) \\
&= round(x_t + \phi_{t-2} + x_{t-1} - y_{t-1}) \\
&= round(x_t + x_{t-1} + \phi_{t-2}) - y_{t-1} \\
&= round(x_t + x_{t-1} + \phi_{t-2}) - round(x_{t-1} + \phi_{t-2}) \\
&= \left(round(\sum_{\tau=1}^{t} x_\tau + \cancel{\phi_0}^{0}) - \sum_{\tau=0}^{t-2} y_\tau\right) - \left(round(\sum_{\tau=1}^{t-1} x_\tau + \cancel{\phi_0}^{0}) - \sum_{\tau=0}^{t-2} y_\tau\right) \\
&= round(\sum_{\tau=1}^{t} x_\tau) - round(\sum_{\tau=1}^{t-1} x_\tau)
\end{aligned} \tag{18}
$$

At which point it is clear that Q is identical to a successive application of a temporal summation, a rounding, and a temporal difference. That is why we say $Q = \Delta \circ R \circ \Sigma$.

# D    UPDATE ALGORITHMS

In Section 2.6, we visually describe what we call the "Past" and "Future" parameters updates. Here we present the algorithms for implementing these schemes.

To simplify our expressions in the update algorithms, we re-parametrize our $k_p, k_d$ coefficients as $k_\alpha \triangleq \frac{k_d}{k_p+k_d}, k_\beta \triangleq \frac{1}{k_p+k_d}$.

$$past : (\bar{x} \in \mathbb{Z}^{d_{in}}, \bar{e} \in \mathbb{Z}^{d_{out}}) \mapsto \widehat{\frac{\partial \mathcal{L}}{\partial w}}_{past}$$

Persistent: $w, u \in \mathbb{R}^{d_{in} \times d_{out}}$,
$$\hat{x} \leftarrow 0^{d_{in}}, \hat{e} \leftarrow 0^{d_{out}}$$
$$i \leftarrow \bar{x} \neq 0, \quad j \leftarrow \bar{e} \neq 0$$
$$\hat{x} \leftarrow k_\alpha \hat{x} \quad , \quad \hat{e} \leftarrow k_\alpha \hat{e}$$
$$v \leftarrow \hat{x}_i \otimes \hat{e}_j \in \mathbb{R}^{\sum_{i'}[\bar{x}_{i'} \neq 0] \times \sum_{j'}[\bar{e}_{j'} \neq 0]}$$
$$\widehat{\frac{\partial \mathcal{L}}{\partial w}}_{past,i,j} \leftarrow \frac{u_{i,j} - v}{1 - k_\alpha^2}$$
$$\hat{x} \leftarrow \hat{x} + k_\beta \bar{x}, \quad \hat{e} \leftarrow \hat{e} + k_\beta \bar{e}$$
$$u_{i,j} \leftarrow v$$

(19)

$$future : (\bar{x} \in \mathbb{Z}^{d_{in}}, \bar{e} \in \mathbb{Z}^{d_{out}}) \mapsto \widehat{\frac{\partial \mathcal{L}}{\partial w}}_{future}$$

Persistent: $w \in \mathbb{R}^{d_{in} \times d_{out}}$,
$$\hat{x} \leftarrow 0^{d_{in}}, \hat{e} \leftarrow 0^{d_{out}}$$
$$\hat{x} \leftarrow k_\alpha \hat{x}$$
$$\hat{e} \leftarrow k_\alpha \hat{e} + k_\beta \bar{e}$$
$$\widehat{\frac{\partial \mathcal{L}}{\partial w}}_{future} \leftarrow \frac{\bar{x} \otimes \hat{e} + \hat{x} \otimes \bar{e}}{k_\alpha^2 - 1}$$
$$\hat{x} \leftarrow \hat{x} + k_\beta \bar{x}$$

(20)

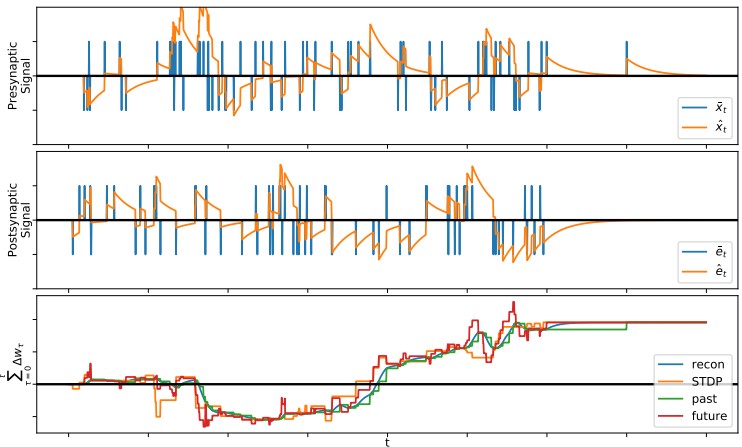

Figure 9: In Section 2.6 and 2.7, we described 4 different update rules ("Reconstruction", "Past", "Future", and "STDP"), and stated that while they do not necessarily produce the same updates at the same times, they produce the same result in the end. Here we demonstrate this empirically. We generate two random spike-trains representing the presynaptic input and the postsynaptic error signal to a single synapse and observe how the weight changes according to the different update rules. **Top**: A randomly generated presynaptic quantized signal $\bar{x}$, along with its reconstruction $\hat{x}$. **Middle**: A randomly generated postsynaptic quantized error signal $\bar{e}$, along with its reconstruction $\hat{e}$. **Bottom**: The cumulative weight update arising from our four updates methods. "*recon*" is just $\sum_{\tau=1}^t \hat{x}_\tau \hat{e}_\tau$, "*past*" and "*future*" are described in Section 2.6 and "*STDP*" is described in Section 2.7. Note that by the end all methods arrive at the same final-weight value.

# E TUNING $k_p$, $k_d$

## E.1 CAUSES OF APPROXIMATION ERROR

Equation 7 shows how we make two approximations when approximating $z_t = x_t \cdot w_t$ with $\hat{z}_t = (dec \circ w \circ Q \circ enc)(x_t)$. The first is the "nonstationary weight" approximation, arising from the fact that w changes in time. The second is the "quantization" approximation, arising from the quantization of x. Here we do a small experiment in which we multiply a time-varying scalar signal $x_t$ with a time-varying weight $w_t$ for many different values of $k_p$, $k_d$ to understand the effects of $k_p$, $k_d$ on our approximation error. The bottom-middle plot in Figure 10 shows that we enter a high-reconstruction-error regime (blue on plot) when $k_d$ is small (high quantization error), or when $k_d >> k_p$ (high nonstationary-weight error). The bottom-right plot shows that blindly increasing $k_p$ and $k_d$ leads to representing the signal with many more spikes. Thus we need to tune hyperparameters to find the "sweet spot" where reconstruction error is fairly low but our encoded signal remains fairly sparse, keeping computational costs low.

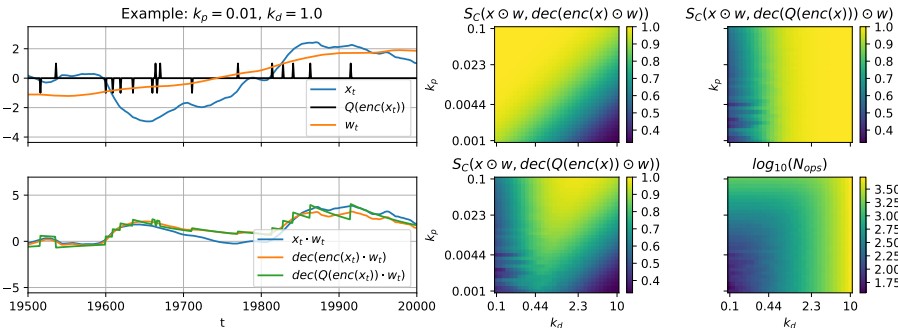

Figure 10: **Top Left**: A time varying signal $x_t$, the quantized signal $Q(enc(x_t))$, and the time-varying "weight" $w_t$. **Bottom Left**: Compare the true product of these signals $x_t \cdot w_t$ with the $dec(enc(x_t \cdot w_t))$, which shows the effects of the non-stationary weight approximation, and $dec(Q(enc(x_t)) \cdot w)$ which shows both approximations. **Top Middle**: The Cosine distance between the "true" signal $x \odot w$ and the approximation due to the nonstationary w, scanned over a grid of $k_p$, $k_d$ values. **Top Right**: The cosine distance between the "true" signal and the approximation due to the quantization of x. **Bottom Middle**: The Cosine Distance between the "true" signal and the full approximation described in Equation 7. This shows why we need both $k_p$ and $k_d$ to be nonzero. **Bottom Right**: The Number of spikes in the encoded signal. In a neural network this would correspond to the number of weight-lookups required to compute the next layer's activation: $dec(Q(enc(x)) \odot w)$.

## E.2 AN AUTO-TUNING SCHEME FOR $k_p$, $k_d$

The smaller the magnitude of a signal, the more severely distorted it is by our quantization-reconstruction scheme. We can see that scaling a signal by K has the same effect on the quantized version of the signal, $s_t$, as scaling $k_p$ and $k_d$ by K: $s_t = (Q \circ enc_{k_p,k_d})(Kx_t) = Q(k_p Kx_t + k_d(Kx_t - Kx_{t-1})) = Q(Kk_p x_t + Kk_d(x_t - x_{t-1})) = (Q \circ enc_{Kk_p,Kk_d})(x_t)$. The fact that the reconstruction quality depends on the signal magnitude presents a problem when training our network, because the error gradients tend to change in magnitude throughout training (they start large, and become smaller as the network learns). To keep our signal within the useful dynamic range of the quantizer, we apply a simple scheme to heuristically adjust $k_p$ and $k_d$ for the forward and backward passes separately, for each layer of the network. Instead of directly setting $k_p$, $k_d$ as hyperparameters, we fix the ratio $k_\alpha \triangleq \frac{k_d}{k_p+k_d}$, and adapt the scale $k_\beta \triangleq \frac{1}{k_p+k_d}$ to the magnitude of the signal. Our update rule for $k_\beta$ is:

$$\mu_t = (1 - \eta_k)\mu_{t-1} + \eta_k \cdot |x_t|_{L_1}$$
$$k_\beta = k_\beta + \eta_k(k_\beta^{rel} \cdot \mu_t - k_\beta) \tag{21}$$

Where $\eta_k$ is the scale-adaptation learning rate, $\mu_t$ is a rolling average of the $L_1$ magnitude of signal $x_t$, and $k_\beta^{rel}$ defines how coarse our quantization should be relative to the signal magnitude (higher means coarser). We can recover $k_p, k_d$ for use in the encoders and decoders as $k_p = (1 - k_\alpha)/k_\beta$ and $k_d = k_\alpha/k_\beta$. In our experiments, we choose $\eta_k = 0.001, k_\beta^{rel} = 0.91, k_{alpha} = 0.91$, and initialize $\mu_0 = 1$.

## F  MNIST RESULTS

Here we show training scores and computation for the PDNN and MLP under different input-orderings (the unordered MNIST vs the ordered Temporal MNIST) and hidden layer depths. We notice no dropoff in performance of the PDNN (as compared to an MLP) with the same architecture as we add hidden layers - indicating that the accumulation of quantization noise over layers appears not to be a problem. For all experiments, the PDNN started with $k_\alpha = 0.5$, and this was increased to $k_\alpha = 0.9$ after 1 epoch (see Appendix A for the meaning of $k_\alpha$). Note that the numbers for Mean Computation are counting additions for the PDNN, and multiply-adds for the MLP, so they are not directly comparable (a 32-bit multiply, if implemented in fixed point, is 32 times more energetically expensive than an add (Horowitz, 2014))

| dataset | hidden_sizes | Network | kOps/sample | Training Score | Test Score |
|---|---|---|---|---|---|
| mnist | [200] | PDNN | 711000 | 100 | 98.34 |
| mnist | [200] | MLP | 314000 | 100 | 98.3 |
| mnist | [200, 200] | PDNN | 1000000 | 99.82167 | 98.18 |
| mnist | [200, 200] | MLP | 434000 | 100 | 98.5 |
| mnist | [200, 200, 200] | PDNN | 1300000 | 99.91 | 98.16 |
| mnist | [200, 200, 200] | MLP | 554000 | 99.99333 | 98.39 |
| mnist | [200, 200, 200, 200] | PDNN | 1620000 | 99.96 | 98.41 |
| mnist | [200, 200, 200, 200] | MLP | 674000 | 99.99167 | 98.28 |
| temporal_mnist | [200] | PDNN | 484000 | 100 | 98.39 |
| temporal_mnist | [200] | MLP | 314000 | 100 | 98.2 |
| temporal_mnist | [200, 200] | PDNN | 740000 | 99.97833 | 98.27 |
| temporal_mnist | [200, 200] | MLP | 434000 | 100 | 98.38 |
| temporal_mnist | [200, 200, 200] | PDNN | 967000 | 99.98 | 98.31 |
| temporal_mnist | [200, 200, 200] | MLP | 554000 | 99.99833 | 98.45 |
| temporal_mnist | [200, 200, 200, 200] | PDNN | 1170000 | 99.995 | 98.18 |
| temporal_mnist | [200, 200, 200, 200] | MLP | 674000 | 100 | 98.53 |

## G    SAMPLE FRAMES FROM THE YOUTUBE-BB DATASET

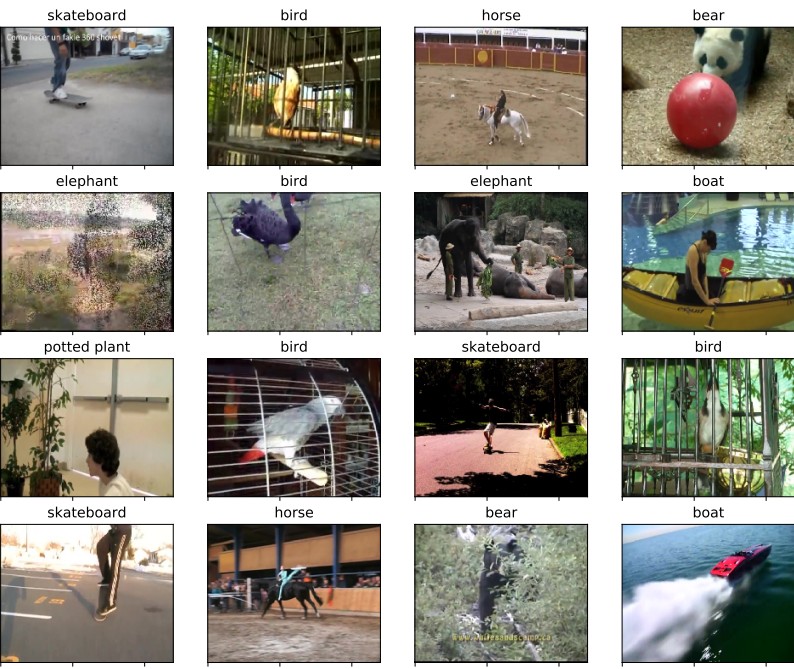

Figure 11: 16 Frames from the Youtube-BB dataset. Each video annotated as having one of 24 objects in it. It also comes with annotated bounding-boxes, which we do not use in this study.

## H    INSTABILITY IN NEURAL NETWORK REPRESENTATIONS

Figure 8 seems to show that computation doesn't quite approach zero as our frame-rate increases, but flat-lines at a certain point. We think this may have to do with the fact that hidden layer activations are not necessarily smoother in time than the input. We demonstrate this by taking 5 video snippets from the Youtube-BB dataset and running them through a (non-spiking) 19 layer VGGNet architectures (Simonyan and Zisserman, 2014), which was pre-trained on ImageNet.

Given these 5 snippets, we measures how much the average relative change in layer activation $\frac{|a_t - a_{t-1}|}{2(|a_t| + |a_{t-1}|)}$ varies as we increase our frame-rate, at various layer-depths. We simulate lower frame rates by skipping every N'th frame of video. (so for example to get a 10FPS frame rate we simply select every 3rd frame of the 30FPS video). For each selected frame rate, and for the given layers, we measure the average inter-frame change at various layers:

$$FPS(n) = 30/n \qquad \text{x-axis} \qquad (22)$$

$$RelChange(n) = \frac{1}{S} \sum_{s=1}^{S} \sum_{t=1}^{T/n} \frac{|a_{nt} - a_{(n-1)t}|}{2(|a_{nt}| + |a_{(n-1)t}|)} \qquad \text{y-axis} \qquad (23)$$

Where:
$S = 5$ is the number of video snippets we average over
$T$ is the number of frames in each snippet
$a_t$ is the activation of a layer at time t
$n$ is the number of frames we are skipping over.

This shows something interesting. While our deeper layers do indeed show less relative change in activation over frames than our input/shallow layers, we note that as frame-rate increases, this seems to approach zero *much more slowly* than our input/shallow layers. This is a problem for our

method, which relies on temporal smoothness in all layers (especially those hidden layers with large amounts of downstream units) to save computation. It suggests that methods for learning *slow feature detectors* - layers that are trained specifically to look for slowly varying features of the input, may be helpful to us.

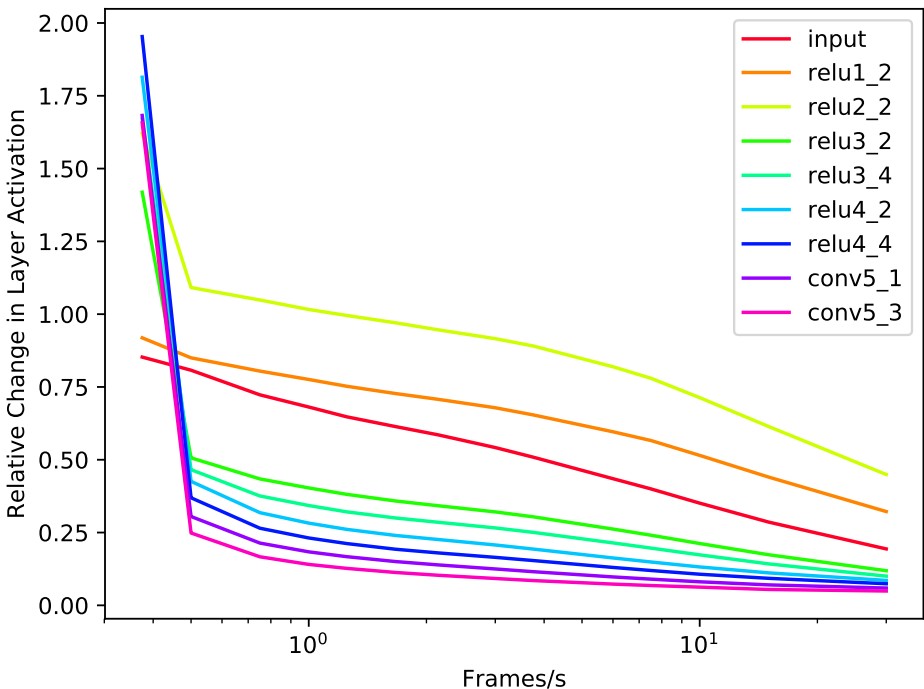

Figure 12: The average relative change in layer activation between frames, as frame-rate increases. For increasing network depth (red=shallow, violet=deep)

