# OpenReview forum: "Temporally Efficient Deep Learning with Spikes"
_ICLR.cc/2018/Conference — Accept (Poster)_

### Official Review · AnonReviewer2 · 2017-11-27
**The paper describes a neural coding scheme for spike based learning in deep neural networks.**

**Rating:** 7
**Confidence:** 5

**Review:**

The principal problem that the paper addresses is how to integrate error-backpropagation learning in a network of spiking neurons that use a form of sigma-delta coding. The main observation is that static sigma-delta coding as proposed in OConnor and Welling (2016b), is not correct when the weights change during training, as past activations are taken into account with the old rather than the new weights.

The solution proposed in this work is to have past activations decay exponentially, to reduce this problem. The coding scheme then mimics the proporitional-integral-derivative idea from control-theory. The result, spikes having an exponentially decaying effect on the postsynaptic neuron, is similar to that observed in biological spiking neurons.

The authors show how spike-based learning can be implemented with spiking neurons using such coding, and demonstrate the results on an MLP with one hidden layer applied to the temporal MNIST dataset, and to the Youtube-BB dataset.

This approach is original and significant, though the presented results are a bit on the thin side. As presented, the spiking networks are not exactly "deep": I am puzzled by the statement that in the youtube-bb dataset only the top 3 layers are "spiking". The network for the MNIST dataset is similarly only 3 layers deep (input, hidden, output). Is there a particular reason for this?  The presentation right now suggests that the scheme does in practise not work for deep networks...

With regard to the learning rule: while the rule is formulated in terms of spikes, it should be noted that for neuron with many inputs and outputs, this update will have to be computed very very often, even for networks with low average firing rates.

The paper is clear in most points, with some parts that could use further elucidation. In particular, in Sec 2.5 the feedback pass for weight updating is computed. It is unclear from the text that this is an ongoing process, in parallel to the feedforward pass. In Sec 2.6 e_t is termed the postsynaptic (pre-nonlinearity) activation, which is confusing as the computation is going the other way (post-to-pre). These two sections would benefit from a more careful layout of the process, what is going on in a forward pass, a backward pass, how does this interact.

Section 2.7 tries to relate the spike-based learning rule to the biologically observed STDP phenomenon. While the formulation in terms of pre-post spike-times is interesting, the result is clearly different from STDP, and ignores the fact that e_t refers to the backpropagating error (which presumably would be conveyed by a feedback network): applying the plotted pre-post spike-time rule in the same setting as where STDP is observed will not achieve error-backpropagation.

The shorthand notation in the paper is hard to follow in the first place btw, perhaps this could be elaborated/remedied in an appendix, there is also some rather colloquial writing in places: "obscene wast of energy" (abstract), "There's" "aren't" (2.6, p5).

The correspondence of spiking neurons to sigma-delta modulation is incorrectly attributed to Zambrano and Bohte (2016), but is rather presented in Yoon (2017/2016, check original date of publication!).

---

### Official Review · AnonReviewer3 · 2017-11-27
**Spike based learning for temporal redundant data**

**Rating:** 6
**Confidence:** 4

**Review:**

This paper presents a novel method for spike based learning that aims at reducing the needed computation during learning and testing when classifying temporal redundant data. This approach extends the method presented on Arxiv on Sigma delta quantized networks (Peter O’Connor and Max Welling. Sigma delta quantized networks. arXiv preprint arXiv:1611.02024, 2016b.). Overall, the paper is interesting and promising; only a few works tackle the problem of learning with spikes showing the potential advantages of such form of computing. The paper, however, is not flawless. The authors demonstrate the method on just two datasets, and effectively they show results of training only for Feed-Forward Neural Nets (the authors claim that “the entire spiking network end-to-end works” referring to their pre-trained VGG19, but this paper presents only training for the three top layers). Furthermore, even if suitable datasets are not available, the authors could have chosen to train different architectures. The first dataset is the well-known benchmark MNIST also presented in a customized Temporal-MNIST. Although it is a common base-line, some choices are not clear: why using a FFNN instead that a CNN which performs better on this dataset; how data is presented in terms of temporal series – this applies to the Temporal MNIST too; why performances for Temporal MNIST – which should be a more suitable dataset — are worse than for the standard MNIST; what is the meaning of the right column of Figure 5 since it’s just a linear combination of the GOps results. For the second dataset, some points are not clear too: why the labels and the pictures seem not to match (in appendix E); why there are more training iterations with spikes w.r.t. the not-spiking case. Overall, the paper is mathematically sound, except for the “future updates” meaning which probably deserves a clearer explanation. Moreover, I don’t see why the learning rule equations (14-15) are described in the appendix, while they are referred constantly in the main text. The final impression is that the problem of the dynamical range of the hidden layer activations is not fully resolved by the empirical solution described in Appendix D: perhaps this problem affects CCNs more than FFN.
Finally, there are some minor issues here and there (the authors show quite some lack of attention for just 7 pages):
-	Two times “get” in “we get get a decoding scheme” in the introduction;
-	Two times “update” in “our true update update as” in Sec. 2.6;
-	Pag3 correct the capital S in 2.3.1
-	Pag4 Figure 1 increase font size (also for Figure2); close bracket after Equation 3; N (number of spikes) is not defined
-	Pag5 “one-hot” or “onehot”;
-	in the inline equation the sum goes from n=1 to S, while in eq.(8) it goes from n=1 to N;
-	Eq(10)(11)(12) and some lines have a typo (a \cdot) just before some of the ws;
-	Pag6 k_{beta} is not defined in the main text;
-	Pag7 there are two “so that” in 3.1; capital letter “It used 32x10^12..”; beside, here, why do not report the difference in computation w.r.t. not-spiking nets?
-	Pag7 in 3.2 “discussed in 1” is section 1?
-	Pag14 Appendix E, why the labels don’t match the pictures;
-	Pag14 Appendix F, explain better the architecture used for this experiment.

---

### Official Review · AnonReviewer1 · 2017-12-03
**Synthesis of deep learning, Sigma-Delta and predictive coding**

**Rating:** 8
**Confidence:** 4

**Review:**

This paper applies a predictive coding version of the Sigma-Delta encoding scheme to reduce a computational load on a deep learning network. Whereas neither of these components are new, to my knowledge, nobody has combined all three of them previously. The paper is generally clearly written and represents a valuable contribution. The authors may want to consider the following comments:

1. I did not really understand the analogy with STDP in neuroscience because it relies on the assumption that spiking of the post-synaptic neuron encodes the backpropagating error signal. I am not aware of any evidence for this. Given that the authors’ algorithm does not reproduce the sign-flip in the STDP rule I would suggest revise the corresponding part of the paper. Certainly, the claim in the Discussion “show these to be equivalent to a form of STDP – a learning rule first observed in neuroscience.” is inappropriate.

2.  If the authors’ encoding scheme really works I feel that they could beef up their experimental results to demonstrate its unqualified advantage.

3. The paper could benefit greatly from better integration with the existing literature.
a. Sigma-Delta model of spiking neurons has a long history in neuroscience starting with the work of Shin. Please note that these papers are much older than the ones you cite:
Shin, J., Adaptive noise shaping neural spike encoding and decoding. Neurocomputing, 2001. 38-40: p. 369-381.
Shin, J., The noise shaping neural coding hypothesis: a brief history and physiological implications. Neurocomputing, 2002. 44: p. 167-175.
Shin, J.H., Adaptation in spiking neurons based on the noise shaping neural coding hypothesis. Neural Networks, 2001. 14(6-7): p. 907-919.
More recently, the noise-shaping hypothesis has been tested with physiological data:
Chklovskii, D. B., & Soudry, D. (2012). Neuronal spike generation mechanism as an oversampling, noise-shaping a-to-d converter. In Advances in Neural Information Processing Systems (pp. 503-511). (see Figure 5A for the circuit implementing a Predictive Sigma-Delta encoder discussed by you)

b. It is more appropriate to refer to encoding a combination of the current value and the increment as a version of predictive coding in signal processing rather than the proportional derivative scheme in control theory because the objective here is encoding, not control. Also, predictive coding has been commonly used in neuroscience:
Srinivasan MV, Laughlin SB, Dubs A (1982) Predictive coding: a fresh view of inhibition in the retina. Proc R Soc Lond B Biol Sci 216: 427–459. pmid:6129637
Using leaky neurons for encoding and decoding is standard, see e.g.:
Bharioke, Arjun, and Dmitri B. Chklovskii. "Automatic adaptation to fast input changes in a time-invariant neural circuit." PLoS computational biology 11.8 (2015): e1004315.
For the application of these ideas to spiking neurons including learning please see a recent paper:
Denève, Sophie, Alireza Alemi, and Ralph Bourdoukan. "The brain as an efficient and robust adaptive learner." Neuron 94.5 (2017): 969-977.

Minor:
Penultimate paragraph of the introduction section: “get get” -> get
First paragraph of the experiments section: ”so that so that” -> so that

---

### Author Response · Authors · 2018-01-03
**Response to Reviewers**

Dear Reviewers,

Thank you for taking the time to read our paper in detail.  Your feedback was very helpful to improving this work.  In response to your suggestions, we have made the following changes to the paper:

- We have reworked the Related Work section, as well as parts of the Abstract and Methods sections, and added Section B of the appendix: Relation to Predictive Coding, to make it clear that our algorithm makes use predictive coding.  We’ve done the same for Shin’s work positing that neurons perform noise-shaping, of which sigma-delta modulation is an instance.

- We’ve added additional explanations for clarity wherever they were asked for.We have added Section A of the appendix - which contains a dictionary of the various notations used throughout the paper.

- We’ve toned down our claim on STDP, making clear that the rule is based on the temporal difference between presynaptic forward pass spikes and postsynaptic backwards pass spikes.

- In response to reviewer R2 asking about training deeper networks, and also in response to the general request for a more extensive experimental analysis, we add a table of results in Appendix G training deeper and deeper networks on MNIST. The main conclusion is that there are no problems with training deeper networks. For the YTBB dataset we follow common practice in computer vision where only the last 3 layers need training when the lower layers have converged. There are not theoretical limitations, however, and assuming a GPU with large enough on-chip memory, the whole network can also be trained.

- We’ve updated figures to make them more readable, and corrected a mistake wherein the one of the Youtube-BB learning curves was not completely plotted, as well as the figure in the appendix where labels were not matched to the images.

- We’ve corrected all the small mistakes you pointed out - thank you for those.

---

### Author Response · Authors · 2018-05-07
**Link to Poster**

https://drive.google.com/file/d/1QJQkZNhdV_hVmQxDOVRxnIk0S-t1g3_p/view?usp=sharing

---

### Decision · Program_Chairs · 2018-01-29
**ICLR 2018 Conference Acceptance Decision**

**Decision:**

Accept (Poster)

**Comment:**

This paper provides an interesting synthesis of ideas. Although the results could be improved, this is a good paper.